# Between-day reliability of local and global muscle-tendon unit assessments in female athletes whilst standardising menstrual cycle phase

Scott Newbould[1]*, Josh Walker[1], Alexander J. Dinsdale[1], Sarah Whitehead[1,2], Gareth Nicholson[1]

**1** Carnegie School of Sport, Leeds Beckett University, Leeds, United Kingdom, **2** Leeds Rhinos Netball, Leeds, United Kingdom

* s.newbould@leedsbeckett.ac.uk

## Abstract

Muscle-tendon unit (MTU) assessments can be categorised into local (e.g., tendon strain) or global (e.g., jump height) assessments. Although menstrual cycle phase may be a key consideration when implementing these assessments in female athletes, the reliability of many MTU assessments is not well defined within female populations. Therefore, the purpose of this study was to report the test-retest reliability of local and global MTU assessments during the early follicular phase of the menstrual cycle. Seventeen naturally menstruating females (age 28.5±7.3 years) completed local and global MTU assessments during two testing sessions separated over 24–72 hours. Local tests included Achilles' tendon mechanical testing and isometric strength of ankle plantar flexors and knee extensors, whereas global tests included countermovement, squat, and drop jumps, and the isometric midthigh pull. Based on intraclass correlation coefficient (ICC) statistics, *poor* to *excellent* reliability was found for local measures (ICC: 0.096–0.936). *Good* to *excellent* reliability was found for all global measures (ICC: 0.788–0.985), excluding the eccentric utilisation ratio (ICC 0.738) and most rate of force development metrics (ICC: 0.635–0.912). Isometric midthigh pull peak force displayed *excellent* reliability (ICC: 0.966), whereas force-time metrics ranged from *moderate* to *excellent* (ICC: 0.635–0.970). Excluding rate of force development (coefficient of variation [CV]: 10.6–35.9%), global measures (CV: 1.6–12.9%) were more reproducible than local measures (CV: 3.6–64.5%). However, local metrics directly measure specific properties of the MTU, and therefore provide valuable information despite lower reproducibility. The novel data reported here provides insight into the natural variability of MTU assessments within female athletes which can be used to enhance the interpretation of other female athlete data, especially that which aims to investigate other aspects of variability, such as the menstrual cycle.

**Data availability statement:** The data supporting this work is freely available on the public repository figshare, DOI: https://doi.org/10.6084/m9.figshare.26477206.v1.

**Funding:** The author(s) received no specific funding for this work.

**Competing interests:** The authors have declared that no competing interests exist.

## Introduction

Measures of muscle-tendon unit (MTU) function are of interest in sport science because of their relationships with athletic performance tasks [1–3]. Sport scientists frequently use local and global MTU assessments to gain insight into these characteristics and to inform training practices. Local assessments are those that directly measure a specific MTU property (e.g., tendon mechanical properties [i.e., strain or stiffness]), whilst global tests focus on movement outcomes, such as jump height or reactive strength. Global measures are underpinned by the aspects of MTU properties measured in the local assessments, for example, rate of force development (RFD) and jump height are influenced by tendon stiffness [4]. Thus, gaining further insight into the interrelationships between local MTU measures and global outcomes can help inform training priorities. Reliably assessing both local and global MTU properties is therefore of great importance to sport scientists. However, much of the sport science literature does not reflect the increasing participation in female sport seen in recent years [5] and there is currently limited data on the sources of variability in MTU assessments in females.

The scarcity of female-specific reliability data is a concern because differences in the hormonal environment between males and females might result in different magnitudes of biological variation within and between days [6]. Specifically, it must be considered that females present with comparatively higher and more variable levels of hormones such as oestrogen, which can impact both neuromuscular output [7] and tendon mechanical properties [8]. However, the true effect of changes in the hormonal environment on MTU function remains unknown [9]. Until these effects are better understood, reliability data should not be directly transferred between sexes, and when collecting female-specific data, consideration should be made for the hormonal environment at the time of testing. Understanding the natural variability of MTU assessments without the confounding effects of hormonal changes would allow more objective inferences to be made regarding MTU testing and monitoring data in female athletes, especially when comparing different hormonal profiles (e.g., different phases of the menstrual cycle).

Regarding local MTU assessments, the triceps surae muscle group and the Achilles' tendon (AT) comprise one of the most profiled muscle-tendon complexes in athletes. This is because of the role that the AT has in overall MTU strategy [10,11], and the relationships that exist between AT mechanical properties, such as elongation and stiffness, and performance in sprint [12], jumping [13], and long-distance running tasks [14]. However, if this field of research is to progress to individualised monitoring and prescription of MTU characteristics [15], it is important to first understand the reliability and sensitivity to change of AT metrics. To this end, several studies have reported test-retest reliability data, with a wide range of intraclass correlation coefficients (ICC) and coefficient of variations (CV) reported for both maximal AT elongation (ICC: 0.56–0.95, CV: 4.1–12.3% [16–20]) and AT stiffness (ICC: 0.64–0.98, CV: 4.7–13.5% [13,16–18,20–23]). This variation in reliability is contributed to by the between-study heterogeneity in methods, such as differences in joint configuration, the exact tendinous structure measured, the region used for the calculation

of stiffness, and the loading protocol used. Importantly, to the authors' knowledge, only one of these studies has included female participants (7 males and 4 females in Hansen et al. [16]), and therefore the reliability of these measures within female populations is not well defined.

For global MTU tests, force plate analyses of the countermovement jump (CMJ), squat jump (SJ), drop jump (DJ), and the isometric mid-thigh pull (IMTP) are frequently used to assess and monitor athletes [e.g., 24,25]. Although recommendations have been made for data collection and analysis procedures for these tests [26–28], the interpretation of these metrics is limited without coinciding reliability data. The reliability of both basic (e.g., jump height, peak force) and more complex CMJ metrics (e.g., RFD) have been investigated within female populations in studies by Moir et al. [29] and Keogh et al. [30], with both studies reporting high reliability for nearly all metrics (ICC: 0.80–0.97). However, reliability data has scarcely been reported in adult females for the squat jump, drop jump, or isometric midthigh pull, reducing the ability to interpret these tests in this population. Therefore, research which provides reliability data for a variety of global MTU assessments in a female population will provide novel and valuable information for researchers and practitioners.

Many studies have aimed to measure the variability in performance over the course of the menstrual cycle [9], although true understanding of menstrual cycle variability cannot be complete without first understanding the natural variability that exists within this population. Therefore, the assessment of variability without potentially confounding changes in the hormonal environment will provide meaningful context in which both past and future female athlete research can be interpreted. Thus, the aim of this study was to report the test-retest reliability of local and global MTU assessments in naturally menstruating female athletes, whilst standardising menstrual cycle phase.

## Materials and methods

### Study design

This study used a within-subject repeated measures design, whereby participants completed two testing sessions separated over 24–72 hours during the early follicular phase of the menstrual cycle (days 1–5). The early follicular phase was chosen because it is the phase with the most stable hormonal profile [31], and the timing of this phase was predicted by calendar-based counting using previous cycle length data and confirmed by the presence of menstrual bleeding [32].

### Participants

Seventeen naturally menstruating females volunteered to participate in this study (age 28.5±7.3 years, stature 166.8±5.6 cm, body mass 62.6±8.2 kg). 'Naturally menstruating' was self-reported and was defined as having cycle lengths between 21–35 days and having had nine or more periods in the previous year [32]. Participants had an average training experience of 8.0±5.9 years and performed 4.8±2.6 training sessions per week, and therefore aligned with Tier 2 of the Participant Classification Framework [33]. Additionally, participants were required to be free from lower-body injuries for six months prior to participation, and to not have a previous AT or patellar tendon rupture. Participant recruitment took place between 6/10/22 and 20/10/23, and all participants provided their written informed consent to participate. The study was approved by the Leeds Beckett Ethics Committee (approval reference 102193) and conformed to the Declaration of Helsinki [34].

### Procedures

All participants completed a familiarisation session approximately seven days prior to the first testing visit, which included submaximal and maximal attempts of each test. For the testing sessions, participants first completed a standardised warm up before completing all tests in the following order: CMJ, SJ, DJ, isometric plantar flexion (including measurements of AT mechanical properties), isometric knee extension, and the IMTP. At least 2 minutes rest was provided between each test. Definitions and calculations for all metrics are provided in S1 and S2 Tables.

**Mechanical properties of the muscle-tendon unit.** All measurements of the triceps surae MTU were performed on the left leg to quasi-randomise for limb dominance. First, resting AT length was measured using the techniques of Barfod et al. [35]. This technique accounts for the curvature of the AT by using flexible measuring tape to measure the distance between the AT origin and insertion. The origin of the AT was defined as the gastrocnemius medialis myotendinous junction (which in turn was defined as the junction between the most distal point of the gastrocnemius medialis and the AT), and the insertion of the AT as the posterior and superior corner of the calcaneus [35]. Participants were then seated on an isokinetic dynamometer (CSMI, Cybex Humac Norm, Stoughton, MA, USA), sampling at a frequency of 1,000 Hz, with a hip joint angle of 120° (180° in anatomical position), knee joint angle of 180° (fully extended), and an ankle joint angle of 90° (neutral). Recording at 15 Hz, a 60 mm, 128-element linear array ultrasound probe (LV7.5/60/128Z-2, 5.0–8.0 MHz; EchoBlaster128 CEXT-1Z, Telemed UAB; Vilnius, Lithuania) was placed over the gastrocnemius medialis myotendinous junction to measure the displacement of this landmark throughout all protocols below. Firstly, the AT moment arm was calculated using the tendon excursion method [36]. Next, participants performed four submaximal and one maximal plantar flexion contraction to precondition the AT [37]. Then, participants performed graded isometric contractions of 5 seconds each with 20 seconds of rest between each contraction, starting at 20 N.m and increasing in steps of 10 N.m until a maximum was reached, with 3–5 maximal attempts being performed [19]. The displacement of the myotendinous junction during each contraction was taken as a measure of AT elongation, which was defined as the difference between the location of the myotendinous junction at rest and during the isometric portion of each contraction. Definitions of each metric of Achilles' tendon mechanical properties are provided in S1 Table. Throughout all dynamometry procedures participants were securely strapped into position to minimise joint rotation. However, a certain amount of joint rotation is inevitable, and this results in misalignment of the ankle joint centre and the axis of rotation of the dynamometer, which causes error in the recorded plantar flexion moments. Whilst this study did not account for this rotation, as reported in the Results section, ankle joint rotation was similar between sessions and thus the error in the plantar flexion moment values will be similar across sessions, leaving the reliability results unaffected. A 2D video camera (Sony RX10 Mk3, Sony Group Corporation, Tokyo, Japan) in video mode (50 Hz) was used to record the lower limb in the sagittal plane, and any observed change in ankle joint angle (i.e., from heel lift and dynamometer movement) was measured. These data were used in conjunction with the tendon excursion data to correct the observed myotendinous junction displacement during the contractions for the displacement due to passive rotation [38].

To investigate the reliability of the manual measurement of AT elongation, a random sample of 10 maximal contractions were selected from the whole dataset. For intra-rater reliability, these images were analysed twice by the lead researcher with an intervening period of 48 hours, and for inter-rater reliability, a second experienced ultrasound operator analysed the same videos. The ICCs and CVs for intra- and inter-rater reliability were 0.994 (95% confidence interval [CI]: 0.976–0.999) and 1.47% (CI: 0.69–2.26%), and 0.936 (CI: 0.762–0.984) and 4.80% (CI: 2.50–7.10%), respectively.

**Vertical jumps.** All vertical jumps were performed on a Kistler force plate (Kistler 9286BA, Switzerland) sampling at 1,000 Hz, and all force data were analysed using BioWare software (version 5.3.0.7, Kistler, Switzerland). The contact threshold for take-off and landing was defined as the time point when force fell below or rose above, respectively, the value equal to 5 standard deviations of the residual force during the final 200 ms of the flight phase [39,40]. Participants completed four successful trials of each jump, with the average of the best three trials used for analysis [41,42]. For the CMJ and SJ, the best three trials were defined as the trials with the greatest flight time-derived jump height (calculated as $d = v_i t + \frac{1}{2} g t^2$, where $d$ = jump height, $V_i$ = initial velocity, and $g$ = acceleration due to gravity, and $t$ = half of flight time), whereas for the DJ, the three trials with the greatest reactive strength index (RSI) were used. For the CMJ and SJ, the reliability of two jump height methods is reported; flight time derived jump height and the jump height calculated using take-off velocity (see S2 Table for definitions and calculations of global MTU metrics). Throughout all jumps, participants were required to keep their hands on their hips to remove the effect of arm swing to ensure jump performance was predominantly driven by the lower limbs [43]. A 30 second rest was provided between each jump trial.

 

**Table 1. Test-retest reliability of local muscle tendon unit metrics.**

| Variable | Session 1 (M±SD) | Session 2 (M±SD) | Change (M±SD) | p | D | ICC (95% CI) | CV (95% CI) | SEM | MDC |
|---|---|---|---|---|---|---|---|---|---|
| Plantar flexion MVC (N.m) | 124.2 (15.9) | 126.2 (20.4) | 2.0 (8.2) | 0.328 | 0.23 | 0.948$^e$ (0.859–0.981) | 3.6 (2.2-5.0) | 4.2 | 11.6 |
| Knee extension MVC (N.m) | 125.1 (30.7) | 130.8 (34.3) | 5.7 (11.5) | 0.079 | 0.46 | 0.961$^e$ (0.875–0.987) | 5.9 (4.0-7.7) | 6.4 | 17.8 |
| Raw elongation (mm) | 16.8 (4.0) | 16.9 (4.3) | 0.15 (2.71) | 0.834 | 0.05 | 0.885$^g$ (0.666–0.960) | 9.0 (5.8-12.2) | 1.4 | 3.9 |
| Passive elongation (mm) | 5.1 (1.3) | 4.9 (1.3) | −0.21 (0.63) | 0.194 | −0.32 | 0.931$^e$ (0.805–0.976) | 7.7 (4.8-10.5) | 0.3 | 0.9 |
| Corrected elongation (mm) | 11.7 (3.2) | 12.0 (3.5) | 0.36 (2.71) | 0.604 | 0.13 | 0.812$^g$ (0.458–0.934) | 13.9 (9.1-18.7) | 1.5 | 4.0 |
| Strain (%) | 6.0 (2.0) | 6.2 (2.0) | 0.16 (1.42) | 0.653 | 0.11 | 0.864$^g$ (0.610–0.953) | 13.9 (9.1-18.7) | 0.7 | 2.1 |
| AT force (N) | 4332.0 (687.8) | 4335.9 (861.9) | 3.9 (477.1) | 0.974 | 0.01 | 0.902$^e$ (0.727–0.965) | 5.8 (3.0-8.6) | 244.1 | 676.6 |
| AT$k_{all}$ (N/m) | 377.9 (143.4) | 324.0 (98.5) | −53.9 (150.8) | 0.173 | −0.34 | 0.497$^p$ (−0.308–0.818) | 20.0 (12.5-27.5) | 96.6 | 267.7 |
| AT$k_{low}$ (N/m) | 223.7 (94.0) | 194.6 (61.3) | −29.1 (81.9) | 0.176 | −0.34 | 0.623$^m$ (−0.012–0.865) | 21.8 (12.1-31.5) | 48.7 | 135.1 |
| AT$k_{high}$ (N/m) | 375.5 (287.7) | 344.7 (357.4) | −30.8 (348.4) | 0.729 | −0.08 | 0.608$^m$ (−0.165–0.865) | 30.9 (−28.7-90.5) | 203.1 | 563.0 |
| NAT$k_{all}$ (N/strain) | 744.2 (263.8) | 615.5 (193.3) | −128.7 (312.0) | 0.120 | −0.39 | 0.096$^p$ (−1.221-0.665) | 22.9 (12.0-33.7) | 213.1 | 590.6 |
| NAT$k_{low}$ (N/strain) | 448.0 (199.1) | 328.4 (309.5) | −119.6 (339.2) | 0.179 | −0.34 | 0.687$^m$ (0.403–0.896) | −43.9 (−166.0-78.1) | 225.38 | 624.71 |
| NAT$k_{high}$ (N/strain) | 732.7 (568.1) | 576.8 (532.7) | −155.9 (559.1) | 0.282 | −0.27 | 0.340$^p$ (−1.295-0.795) | 64.5 (11.3-117.7) | 326.25 | 904.33 |
| AT$k_{index}$ (N/strain) | 780.6 (219.8) | 747.3 (208.0) | −33.3 (172.8) | 0.452 | −0.18 | 0.809$^g$ (0.460–0.933) | 12.4 (7.6-17.1) | 93.5 | 259.2 |
| AT moment arm (mm) | 28.9 (2.9) | 29.5 (3.6) | 0.6 (2.41) | 0.337 | 0.23 | 0.842$^g$ (0.574–0.942) | 4.0 (2.1-5.9) | 1.3 | 3.6 |
| Ankle rotation (°) | 11.0 (2.7) | 10.8 (3.1) | −0.3 (1.3) | 0.415 | −0.19 | 0.946$^e$ (0.853–0.980) | 7.8 (5.2-10.3) | 0.7 | 1.9 |

M, mean; SD, standard deviation; ICC, intraclass correlation coefficient; CV, coefficient of variation; SEM, standard error of measurement; MDC, minimum detectable change; AT, Achilles' tendon; AT$k$, Achilles' tendon stiffness; NAT$k$, normalised Achilles' tendon stiffness. Superscript letters refer to ICC classification, p = poor, m = moderate, g = good, e = excellent. The definition and calculation for each variable is provided in S1 Table.

For the CMJ, participants were instructed to start standing upright, then perform a countermovement to a self-selected depth immediately into a maximal effort vertical jump, aiming to maximise vertical displacement of the centre of mass [26]. For the SJ, participants were instructed to descend to a self-selected squat depth [44], hold this posture for 3 seconds, then perform a maximum effort concentric-only vertical jump, once again aiming to maximise vertical displacement. Any trials with a visible countermovement or that displayed a force unload of ≥10% of body mass before the initiation of the jump were excluded [45]. The onset of both the CMJ and SJ was defined as the time point 30 ms before the vertical force trace moves beyond ± 5 standard deviations of body weight from the quiet weighing period [46]. Finally, for the DJ, participants were instructed to step forwards off a 30 cm box and to perform a maximal effort vertical jump immediately once they make ground contact, whilst keeping ground contact time to a minimum [27].

**Isometric strength tests.** As well as the isometric plantar flexion contractions above, participants also performed unilateral isometric knee extensions and the bilateral IMTP. For both tests, participants performed three submaximal trials (at approximately 25, 50, and 75% of maximum effort) and three maximal effort trials. For the knee extension, participants were seated on the Cybex dynamometer with a knee joint angle of 90° and a hip joint angle of 110°, and all contractions were held for 5 seconds. All participants used their left leg to quasi-randomise for limb dominance. IMTP testing followed the setup and protocol guidelines set out by Comfort et al. [28]. In brief, participants stood in a squat rack on a force plate (Kistler 9253B, Switzerland) with a knee joint angle of 125–145° and hip joint angle of 140–150°, with the bar fixed at mid-thigh level. The participants were then instructed to push their feet into the floor and pull on the bar "as fast and hard as possible" for 3 seconds [47,48]. Onset for the IMTP was defined in the same manner as for the CMJ and SJ [49]. Participants had 30 and 60 seconds of rest between each trial of the KE and IMTP, respectively. Strong verbal encouragement was used throughout all strength tests.

## Statistical analysis

All data are presented as mean ± standard deviation (SD). Data were checked for normality using a Shapiro-Wilk test, and any variables that were not normally distributed were logarithmically transformed before the calculation of ICCs. To investigate relative and absolute test-retest reliability, the ICC (two-way mixed effects, mean of $k$ measurements, absolute agreement [50]) and CV were used, respectively, both with 95% CI. The ICC was interpreted as $\leq 0.50 = poor$, 0.50 to $0.75 = moderate$, 0.75 to $0.90 = good$, and $\geq 0.90 = excellent$ [50]. The CV was calculated for each participant as (*within-subject SD / mean*) * 100, with the mean of values from all participants used as the test-retest CV (e.g., [24]). Additionally, the standard error of measurement (SEM, calculated as $SD_{pooled} * \sqrt{(1-ICC)}$) and the minimum detectable change (MDC, calculated as $1.96 * \sqrt{2} * SEM$) were calculated. For intra-rater reliability the ICC form used was a two-way mixed effects, single measurement, absolute agreement, whereas for inter-rater reliability, the two-way random effects, single rater, absolute agreement was used [50]. Finally, a two-tailed paired *t*-test (or Wilcoxon test in the case of non-normally distributed data) was used to assess systematic bias between testing sessions, with $\alpha = 0.05$. Microsoft Excel (Microsoft Corporations, Redmond, WA, USA) and SPSS Statistics (IBM Corporations, Version 28.0, Armonk, NY, USA) were used for calculations and statistical analyses. Due to technical issues, data are missing for one participant for AT elongation, strain, and stiffness measures, and for two participants for the IMTP and knee extension tests.

## Results

All results, including the ICC, CV, mean change, effect size and p-value of change, SEM and MDC, as well as the relevant ICC-based descriptor are displayed for each variable in Tables 1–5. For local MTU metrics, the plantar flexion and knee extension MVCs displayed *excellent* reliability, the AT moment arm and elongation measures all displayed *good* to *excellent* reliability, whereas all stiffness metrics displayed *moderate* or *poor* reliability, except $ATk_{index}$ which showed *good* reliability (Table 1). All CMJ variables displayed *good* to *excellent* reliability (Table 2). For the SJ, all metrics showed *excellent* reliability, except RFD time bands which were all *good*, and the eccentric utilisation ratio, which was *moderate* (Table 3). All DJ variables displayed *excellent* reliability (Table 4). Peak force from the IMTP displayed *excellent* reliability, whereas results for force-time variables were mixed, ranging from *moderate* to *excellent*. Finally, the DSI displayed *good* reliability (Table 5).

## Discussion

The aim of this study was to report the test-retest reliability of local and global MTU assessments in naturally menstruating female athletes, whilst standardising menstrual cycle phase at the time of testing. Local measures, which included metrics of AT mechanical properties and single-joint isometric strength tests, displayed *good* to *excellent* reliability, except for traditional AT stiffness metrics which were *poor* to *moderate*. For global measures, *good* to *excellent* reliability was found

**Table 2. Test-retest reliability of countermovement jump metrics.**

| Variable | Session 1 (M±SD) | Session 2 (M±SD) | Change (M±SD) | p | d | ICC (95% CI) | CV (95% CI) | SEM | MDC |
|---|---|---|---|---|---|---|---|---|---|
| Take-off velocity (m/s) | 2.061 (0.146) | 2.082 (0.149) | 0.021 (0.051) | 0.084 | 0.39 | 0.966$^e$ (0.906–0.988) | 1.6 (1.1-2.1) | 0.027 | 0.075 |
| Jump height from take-off velocity (cm) | 21.75(3.00) | 22.20 (3.10) | 0.40 (1.07) | 0.102 | 0.40 | 0.966$^e$ (0.905–0.988) | 3.1 (2.2-4.1) | 0.57 | 1.59 |
| Flight time (s) | 0.442 (0.033) | 0.448 (0.034) | 0.006 (0.011) | 0.049* | −0.49 | 0.966$^e$ (0.895–0.988) | 1.7 (1.2-2.2) | 0.006 | 0.017 |
| Jump height from flight time (cm) | 24.06(3.50) | 24.70 (3.56) | 0.64 (1.21) | 0.049* | 0.50 | 0.966$^e$ (0.895–0.988) | 3.4 (2.3-4.4) | 0.68 | 1.88 |
| Peak force (N) | 1356.9 (233.2) | 1394.1 (227.6) | 37.2 (126.0) | 0.241 | 0.28 | 0.917$^e$ (0.777–0.970) | 4.6 (2.6-6.6) | 66.4 | 184.0 |
| Peak power (W) | 2386.0 (391.7) | 2416.3 (395.5) | 30.3 (106.1) | 0.256 | 0.27 | 0.981$^e$ (0.949–0.993) | 2.2 (1.2-3.1) | 54.3 | 150.4 |
| Concentric RPD (W/s) | 12495.7 (3688.9) | 12842.8 (4060.1) | 347.2 (3231.4) | 0.943 | 0.10 | 0.788$^g$ (0.402–0.924) | 12.9 (5.7-20.1) | 1743.4 | 4832.4 |
| Rate of force development (N/s) | 3993.6 (1641.1) | 4763.7 (2371.8) | 770.2 (1902.7) | 0.163 | 0.39 | 0.791$^g$ (0.443–0.923) | 17.9 (9.3-26.6) | 1115.2 | 3091.2 |
| Positive impulse (N.s) | 468.2 (80.3) | 450.7 (80.5) | −17.6 (57.2) | 0.653 | −0.29 | 0.875$^g$ (0.665–0.954) | 5.2 (1.7-8.7) | 31.1 | 86.3 |
| Countermovement depth (cm) | 25.93 (5.15) | 25.65 (5.70) | 0.28 (3.19) | 0.718 | −0.09 | 0.910$^e$ (0.750–0.967) | 6.7 (3.6-9.9) | 1.63 | 4.52 |
| Vertical stiffness (N/m) | 5526.3 (1656.6) | 5744.8 (1669.9) | 218.6 (1433.8) | 0.795 | 0.15 | 0.840$^g$ (0.561–0.942) | 10.3 (5.2-15.4) | 783.7 | 2172.2 |
| RSImod | 0.302 (0.064) | 0.319 (0.060) | 0.017 (0.0416) | 0.116 | −0.38 | 0.860$^g$ (0.620–0.949) | 7.2 (4.2-10.2) | 0.023 | 0.064 |

M, mean; SD, standard deviation; ICC, intraclass correlation coefficient; CV, coefficient of variation; SEM, standard error of measurement; MDC, minimum detectable change; RFD, rate of force development; RPD, rate of power development; RSImod, reactive strength index modified. Superscript letters refer to ICC classification, p = poor, m = moderate, g = good, e = excellent. * indicates a significant (p < 0.05) difference between sessions. The definition and calculation for each variable is provided in S2 Table.

for all CMJ metrics, all SJ metrics except eccentric utilisation ratio (*moderate*), and all DJ metrics. IMTP peak force also showed *excellent* reliability, whereas IMTP force-time metrics ranged from *moderate* to *excellent*. An overview of the ICC for each variable is provided in Fig 1. To the authors' knowledge, this is the first study to investigate the reliability of many of these metrics in a female sample, and the first study to standardise menstrual cycle phase of the participants at the time of testing. Therefore, these results provide highly novel insight into the natural variability of MTU assessments within female athletes, data which can be used to enhance the interpretation of a vast array of past and future female athlete research.

A wide range of reliability scores were seen across the local and global measures in this study (Fig 1), and local measures generally had higher CVs than global ones (except for single-joint isometric moments and global RFD metrics). When interpreting these findings, it is important to consider that any change between sessions is comprised of a combination of measurement error and biological variation [51]. It is plausible that any biological variation that is present could directly affect local measures, but only indirectly affect global ones. This is because local metrics directly measure some specific aspect of the MTU (e.g., tendon strain) whereas global measures have more degrees of freedom and may employ many different motor strategies to achieve a consistent performance outcome. Therefore, the performance outcome in a global test can still be maintained from session to session despite potential changes to MTU properties that underpin the movement. For example, there is the potential for jump strategies to differ between sessions, but the overall outcome of

**Table 3. Test-retest reliability of squat jump metrics.**

| Variable | Session 1 (M±SD) | Session 2 (M±SD) | Change (M±SD) | P | d | ICC (95% CI) | CV (95% CI) | SEM | MDC |
|---|---|---|---|---|---|---|---|---|---|
| Take-off velocity (m/s) | 1.984 (0.166) | 1.983 (0.156) | −0.01 (0.080) | 0.949 | −0.02 | 0.937$^e$ (0.826–0.977) | 2.2 (1.3-3.0) | 0.040 | 0.112 |
| Jump height from take-off velocity (cm) | 20.21 (3.23) | 20.16 (3.11) | −0.05 (1.63) | 0.911 | −0.03 | 0.933$^e$ (0.814–0.976) | 4.3 (2.6-6.1) | 0.08 | 2.28 |
| Flight time (s) | 0.420 (0.034) | 0.421 (0.033) | 0.002 (0.014) | 0.619 | 0.11 | 0.962$^e$ (0.895–0.986) | 2.0 (1.5-2.5) | 0.007 | 0.019 |
| Jump height from flight time (cm) | 21.80 (3.40) | 21.90 (3.23) | 0.10 (1.00) | 0.723 | 0.07 | 0.960$^e$ (0.891–0.986) | 4.1 (3.1-5.1) | 0.73 | 2.03 |
| Peak force (N) | 1262.6 (198.4) | 1297.4 (221.7) | 34.8 (69.6) | 0.056 | 0.48 | 0.967$^e$ (0.897–0.988) | 3.1 (1.9-4.3) | 38.2 | 105.9 |
| Peak power (W) | 2305.9 (419.5) | 2327.0 (416.8) | 21.2 (104.6) | 0.416 | 0.19 | 0.984$^e$ (0.958–0.994) | 2.3 (1.4-3.3) | 52.9 | 146.6 |
| RPD (W/s) | 6825.8 (1983.8) | 7485.7 (2464.4) | 659.8 (848.0) | 0.005$^*$ | 0.74 | 0.944$^e$ (0.727–0.983) | 8.4 (5.9-11.0) | 529.4 | 1467.4 |
| RFD (N/s) | 2196.9 (991.4) | 2692.7 (1475.3) | 495.8 (642.1) | 0.010$^*$ | 0.74 | 0.912$^e$ (0.625–0.972) | 16.2 (11.2-21.2) | 403.4 | 1118.1 |
| RFD$_{50}$ (N/s) | 1564.6 (984.6) | 1735.2 (1039.5) | 170.7 (784.1) | 0.383 | 0.21 | 0.825$^g$ (0.526–0.936) | 24.7 (14.8-34.6) | 423.5 | 1173.9 |
| RFD$_{100}$ (N/s) | 2367.4 (1410.3) | 2873.8 (1674.2) | 506.4 (1125.9) | 0.082 | 0.43 | 0.829$^g$ (0.532–0.938) | 26.3 (16.5-36.1) | 640.1 | 1774.2 |
| RFD$_{250}$ (N/s) | 2094.5 (808.8) | 2081.6 (773.9) | −12.9 (632.9) | 0.934 | −0.02 | 0.819$^g$ (0.490–0.935) | 16.4 (9.4-23.4) | 336.7 | 933.4 |
| Eccentric Utilisation Ratio | 1.08 (0.10) | 1.11 (0.08) | 0.02 (0.08) | 0.312 | 0.24 | 0.738$^m$ (0.294–0.904) | 4.4 (2.9-5.8) | 0.05 | 0.13 |

M, mean; SD, standard deviation; ICC, intraclass correlation coefficient; CV, coefficient of variation; SEM, standard error of measurement; MDC, minimum detectable change; RPD, rate of power development; RFD, rate of force development. Superscript letters refer to ICC classification, p = poor, m = moderate, g = good, e = excellent. * indicates a significant (p < 0.05) difference between sessions. The definition and calculation for each variable is provided in S2 Table.

**Table 4. Test-retest reliability of drop jump metrics.**

| Variable | Session 1 (M±SD) | Session 2 (M±SD) | Change (M±SD) | P | d | ICC (95% CI) | CV (95% CI) | SEM | MDC |
|---|---|---|---|---|---|---|---|---|---|
| Flight time (s) | 0.391 (0.056) | 0.402 (0.051) | 0.011 (0.013) | 0.006$^*$ | 0.81 | 0.946$^e$ (0.868–0.993) | 2.6 (1.7-3.6) | 0.008 | 0.023 |
| Jump height from flight time (cm) | 19.16 (4.93) | 20.11 (4.75) | 0.95 (1.15) | 0.006$^*$ | 0.79 | 0.976$^e$ (0.872–0.993) | 5.2 (3.4-7.1) | 0.730 | 2.040 |
| Ground contact time (s) | 0.227 (0.031) | 0.226 (0.028) | −0.001 (0.013) | 0.723 | −0.08 | 0.953$^e$ (0.869–0.983) | 3.1 (2.0-4.2) | 0.006 | 0.018 |
| Reactive strength index | 0.876 (0.277) | 0.915 (0.263) | 0.038 (0.053) | 0.008$^*$ | 0.69 | 0.985$^e$ (0.925–0.995) | 4.8 (2.6-7.0) | 0.032 | 0.089 |
| Peak force (N) | 3204.5 (808.4) | 3145.4 (765.2) | −59.1 (212.1) | 0.268 | −0.27 | 0.981$^e$ (0.949–0.993) | 3.4 (2.2-4.6) | 108.5 | 300.7 |

M, mean; SD, standard deviation; ICC, intraclass correlation coefficient; CV, coefficient of variation; SEM, standard error of measurement; MDC, minimum detectable change. Superscript letters refer to ICC classification, p = poor, m = moderate, g = good, e = excellent. * indicates a significant (p < 0.05) difference between sessions. The definition and calculation for each variable is provided in S2 Table.

jump height does not (e.g., [52]). As such, global measures such as jump height may lack the precision to detect biological changes in MTU function, whereas local measures can provide more sensitive insights into specific aspects of the MTU.

Two local MTU metrics, plantar flexion and knee extension MVCs, displayed *excellent* reliability (Table 1). These single joint isometric actions largely remove the 'skill' aspect of strength testing [53] and provide reliable insights into the

**Table 5. Test-retest reliability of isometric midthigh pull metrics.**

| Variable | Session 1 (M±SD) | Session 2 (M±SD) | Change (M±SD) | P | d | ICC (95% CI) | CV (95% CI) | SEM | MDC |
|---|---|---|---|---|---|---|---|---|---|
| Peak force (N) | 1614.1 (211.9) | 1587.6 (209.9) | −26.5 (73.9) | 0.187 | −0.34 | 0.966$^e$ (0.902–0.989) | 2.8 (1.8-3.7) | 38.9 | 107.8 |
| Force$_{50}$ (N) | 726.3 (89.0) | 716.7 (104.4) | −9.6 (38.7) | 0.353 | −0.24 | 0.959$^e$ (0.881–0.986) | 3.1 (1.8-4.4) | 19.6 | 54.4 |
| Force$_{100}$ (N) | 909.3 (145.5) | 875.4 (148.4) | −33.8 (132.8) | 0.341 | −0.24 | 0.744$^m$ (0.256–0.913) | 7.3 (3.4-11.1) | 74.4 | 206.1 |
| Force$_{250}$ (N) | 1337.2 (247.9) | 1330.4 (218.9) | −6.9 (145.3) | 0.857 | −0.05 | 0.899$^g$ (0.697–0.966) | 6.4 (3.7-9.2) | 74.3 | 206.0 |
| Impulse$_{50}$ (N.s) | 33.6 (3.9) | 32.9 (4.3) | −0.7 (1.3) | 0.051 | −0.53 | 0.970$^e$ (0.893–0.990) | 2.6 (1.6-3.5) | 0.7 | 2.0 |
| Impulse$_{100}$ (N.s) | 74.2 (8.9) | 72.9 (10.2) | −1.3 (4.6) | 0.288 | −0.27 | 0.937$^e$ (0.819–0.979) | 3.5 (1.9-5.1) | 2.4 | 6.7 |
| Impulse$_{250}$ (N.s) | 239.4 (43.5) | 239.6 (39.4) | 0.2 (33.1) | 0.978 | 0.01 | 0.821$^g$ (0.452–0.940) | 7.1 (2.8-11.4) | 17.6 | 48.7 |
| RFD$_{50}$ (N/s) | 1629.8 (1057.3) | 1811.7 (1193.2) | 181.9 (987.9) | 0.488 | 0.17 | 0.768$^g$ (0.313–0.922) | 35.9 (18.2-53.7) | 543.0 | 1505.1 |
| RFD$_{100}$ (N/s) | 2484.1 (1234.5) | 2488.9 (1248.1) | 4.8 (1304.5) | 0.989 | 0.00 | 0.635$^m$ (−0.147–0.879) | 29.0 (11.4-46.6) | 749.9 | 2078.7 |
| RFD$_{250}$ (N/s) | 2816.7 (931.8) | 2821.3 (817.8) | 4.6 (449.0) | 0.696 | 0.01 | 0.934$^e$ (0.802–0.978) | 10.6 (4.4-16.7) | 225.2 | 624.3 |
| Dynamic strength index | 0.866 (0.100) | 0.904 (0.151) | 0.038 (0.105) | 0.183 | 0.34 | 0.788$^g$ (0.398–0.928) | 5.9 (3.3-8.5) | 0.059 | 0.163 |

M, mean; SD, standard deviation; ICC, intraclass correlation coefficient; CV, coefficient of variation; SEM, standard error of measurement; MDC, minimum detectable change; RFD, rate of force development. Superscript letters refer to ICC classification, p = poor, m = moderate, g = good, e = excellent.

The definition and calculation for each variable is provided in S2 Table.

force-generating capacity of a muscle group. The consistent plantar flexion MVC provides a conducive environment for measuring the mechanical properties of the AT, and this analysis found that all AT metrics displayed *good* to *excellent* reliability, except for stiffness measures which all displayed either *moderate* or *poor* reliability (Table 1). In contrast, AT$k_{index}$, calculated as maximum strain divided by maximum AT force [19], displayed *good* reliability. In congruence with the present study, Rogers et al. [23] found that a similar variable (stiffness calculated as maximum torque divided by maximum elongation) was more reliable than traditional stiffness variables (ICC: 0.87), and Hunter et al. [19] also reported *good* reliability for AT$k_{index}$ in males (ICC: 0.75). This metric could be more reliable than traditional stiffness variables (that calculate stiffness over certain ranges of the force-elongation curve) as it only takes into account maximal contractions, meaning that any differences in the lower and mid-portions of the force-elongation curve between days would not influence the outcome. In traditional stiffness variables, these small differences are combined, and possibly amplified, in the subsequent calculations, which may partially explain their lower reliability.

As traditional stiffness metrics account for AT force-elongation properties over a greater range of forces, these metrics might provide a greater fidelity of information, but at the cost of lower between-day reliability. This discrepancy in reliability between different stiffness metrics might also explain the mixed reliability findings from previous research [13,16–18,20–23], as there is considerable between-study heterogeneity in stiffness calculation methods, as well as actual measurement methods (e.g., chosen landmark for tracking, contraction protocol). As this is the one of the first studies to assess the reliability of these metrics in female athletes, the menstrual cycle phase of the participants was standardised to account for one potentially confounding source of error. Whilst it is difficult to distinguish between the contribution of biological variation and measurement error to the variability in these metrics, the manual measurement of AT elongation displayed *excellent* intra- and inter-rater reliability in this study. This eliminates one key aspect of measurement error and increases the likelihood that biological variability is the driver of the observed changes, which means that stiffness metrics are perhaps not unreliable *per se*, but are highly sensitive to small, biologically real changes (e.g., [54]). Ultimately, these metrics provide direct insight into a specific MTU property, and because previous research has highlighted that AT properties are able to distinguish between cohorts [55] and identify changes due to training interventions [56], these metrics may still be of interest to researchers providing that the reproducibility of the metrics is kept in mind.

Regarding global tests, most variables from the CMJ, SJ, and DJ showed *good* to *excellent* reliability, with very little change between sessions (Tables 2–4). Insights into these global measures of lower-body performance (e.g., jump height,

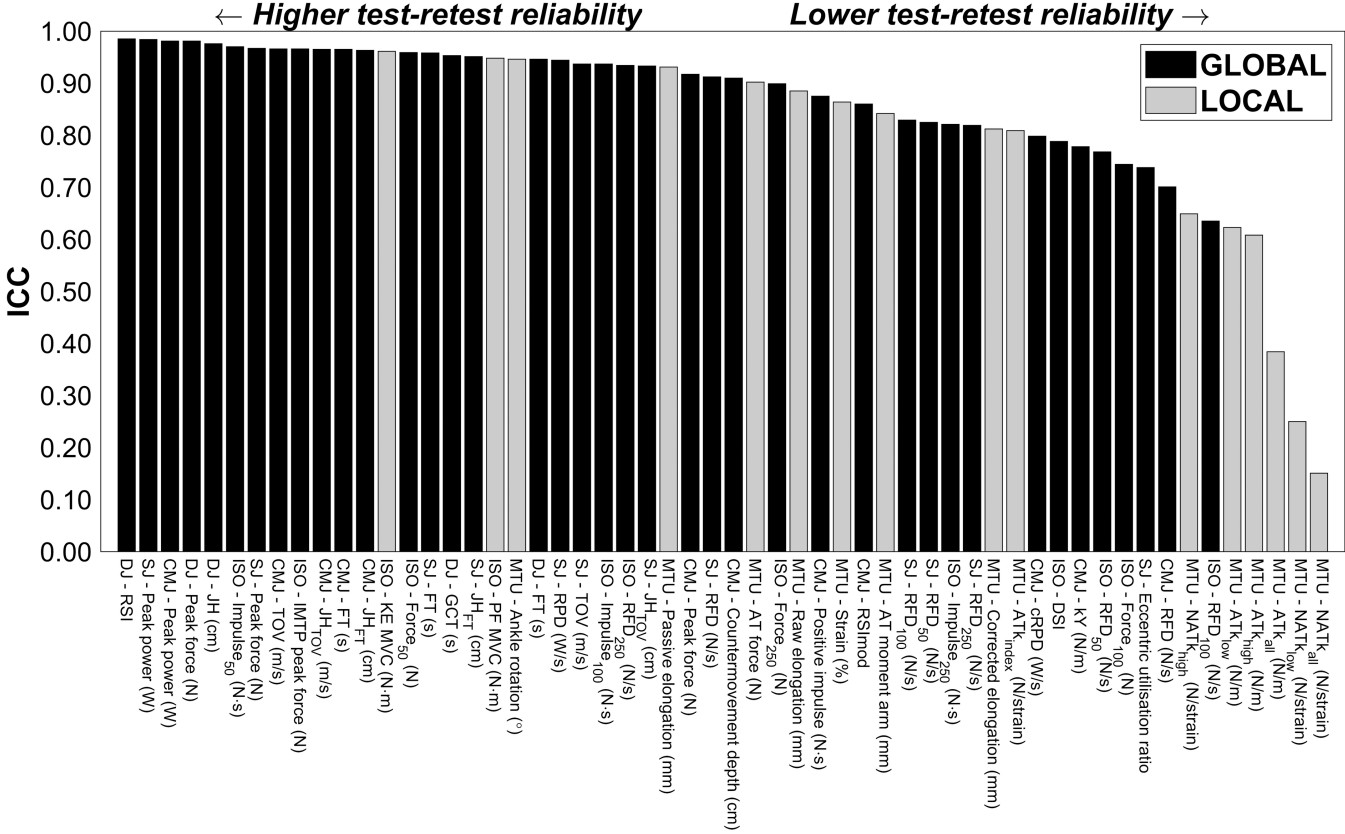

**Fig 1. Intraclass correlation coefficients (ICC) for all metrics.** MTU, muscle-tendon unit; ATk, Achilles' tendon stiffness; NATk normalised Achilles' tendon stiffness; AT$k_{index}$, Achilles' tendon stiffness index; CMJ, countermovement jump; SJ, squat jump; DJ, drop jump; IMTP, isometric midthigh pull; ISO, isometric; RFD, rate of force development; RPD, rate of power development; RSI, reactive strength index; RSImod, reactive strength index modified; kY, vertical stiffness; DSI, dynamic strength index; TOV, take-off velocity; FT, flight time; MVC, maximum voluntary contraction; PF, plantar flexion; KE, knee extension.

force production, or RSI) are useful for measuring adaptations to training interventions, and because of their high reliability, any observed changes are likely meaningful. However, because of their increased degrees of freedom allowing for maintenance of the overall performance outcome, these types of outcomes may not be sensitive enough to detect smaller acute changes such as those expected to occur over menstrual cycle phases [9] or because of fatigue [57]. Furthermore, although MTU properties are often inferred through global tests, it must be acknowledged that only local tests can provide objective evidence of specific MTU properties. For example, RSI and CMJ performance are often used as markers of stretch-shortening cycle function [e.g., 58,59], although evidence has shown that true stretch-shortening cycles do not occur for the triceps surae musculature during the CMJ [10] or DJ [60]. Therefore, as well as considering reliability and sensitivity, practitioners should be mindful of the information that each metric can provide when choosing tests and consider how this might inform training and testing practices. A final consideration for global metrics is the use of composite metrics which are comprised of two other variables, such as the eccentric utilisation ratio being calculated using jump heights from the CMJ and SJ [61] and the dynamic strength index calculated from CMJ and IMTP peak force [62]. Importantly, the results of the present study show that even though the input variables show *excellent* reliability, these composite metrics displayed *moderate* and *good* reliability, respectively. Thus, caution should be taken when considering these types of metrics and what information they are able to provide above their constituent parts.

In this study, although high reliability was found for nearly all vertical jump metrics, systematic bias was present between testing sessions for CMJ and DJ flight time and its associated jump height, DJ RSI, and SJ RFD and rate of power development (RPD). The values for these variables were greater in session two, and despite this bias, all variables displayed *excellent* ICCs and CVs < 10% (excluding SJ RFD), potentially highlighting that the bias is reflecting a small, but real change between sessions. This change might be caused by changes in exercise technique (e.g., changes in landing posture influencing flight time in the CMJ and DJ), or through participants developing the neuromuscular skill of the movement (i.e., a learning effect). This is particularly salient for the DJ, which because of its technical complexity, has been shown to be less reliable than the CMJ and SJ [63]. Thus, the technique and skill requirements for these global metrics and the potential for bias must be considered when choosing tests. Familiarisation sessions can be used to reduce the risk of bias due to a learning effect, and in the present study a familiarisation session was completed by all participants despite previous research showing that familiarisation is not necessary for these three jump tests [64–66]. As the results presented here suggest that more familiarisation sessions are required, future research should explore the effects of multiple familiarisations within a trained female athlete sample. Furthermore, CMJ and SJ depths were not controlled in this study which presents another potential source of variability, and therefore future research should also explore the effects of controlling these variables on the reliability of all jump metrics. Overall, due to their generally *excellent* reliability, once participants are sufficiently familiarised with the movement (to negate systematic bias), these vertical jump tests can reliably assess lower-body performance capacity.

Much has been written about the importance of muscular strength and RFD for sports performance [1]. Muscular strength, measured either locally via plantar flexion and knee extension MVCs or globally via IMTP peak force, showed *excellent* reliability, underscoring the high reliability of strength tests with limited degrees of freedom (Tables 1 and 5). The RFD measures from the CMJ, SJ and IMTP resulted in *good* to *excellent* ICCs, but also high CVs (10.6–35.9%). In the SJ and IMTP, where RFD was measured at 50, 100, and 250 ms from onset, the 250 ms time band proved to be the most reliable. It is expected that calculating RFD over longer time epochs may result in improved reliability, as the absolute change in force measurement produces a smaller relative error over longer time periods and greater force outputs, a trend which has been reported in previous studies [49,67,68]. Additionally, early RFD is driven largely by neuromuscular contributions [47] and has been shown to be more responsive to training [2], highlighting that it may be more sensitive to change than the later time bands, explaining its poorer reliability scores. Therefore, if RFD is to be used, whilst longer time bands (i.e., 250 ms) provide better reliability, it should be considered which aspects of neuromusculoskeletal function are of interest when choosing metrics. It should also be noted that for the IMTP, poorer reliability was found for RFD than the other IMTP force-time metrics (force and impulse over the same 50, 100, and 250 ms time bands as RFD; Table 5), which is in accordance with previous research [67,69,70]. Similarly, RPD displayed greater reliability than RFD in the CMJ and SJ both in the present study and previous investigations [43,71]. Therefore, these findings caution against the use of RFD and promote the use of other force-time metrics instead, such as RPD or impulse.

This study standardised menstrual cycle phase using best practice recommendations [32] to understand the natural variability of the chosen metrics in female athletes without the potentially confounding influence of menstrual cycle related hormone changes. Although this design provides a suitable level of control, future research should measure serum hormone concentrations and record menstrual cycle symptoms to identify the effects that these factors have on outcomes. In addition, to provide a greater scope of support to female athletes, research should also investigate the variability of measures within individuals with other hormonal profiles, such as hormonal contraceptive users or post-menopausal athletes.

## Conclusions

This study investigated the test-retest reliability of local and global assessments in female athletes, with the reliability of many variables reported for the first time. This is the first reliability study investigating MTU, strength, and jump data to standardise the menstrual cycle phase of the participants at the time of testing. In summary, local metrics were generally

less reproducible than global metrics. However, these metrics allow measurement of specific MTU characteristics that influence function, and as such, can provide valuable information. *Good* to *excellent* reliability was found for single-joint isometric strength and metrics of AT mechanical properties, except for traditional stiffness metrics, which were *poor* to *moderate*. On the other hand, global metrics generally displayed high reliability, with *good* to *excellent* ICCs reported for almost all CMJ, SJ, DJ, and IMTP metrics. These tests can reliably assess overall performance outcomes; however, their increased degrees of freedom means that they lack the necessary precision to monitor and infer specific aspects of MTU function. The data reported here provides insight into the natural variability of MTU assessments within a female athlete population, and therefore this data can be used to contextualise and enhance the interpretation of other female athlete data, particularly that which investigates other aspects of variability, such as the menstrual cycle.

## Supporting information

**S1 Table. Definitions and calculations of local muscle-tendon unit metrics.**
(DOCX)

**S2 Table. Definitions and calculations of global muscle-tendon unit metrics.**
(DOCX)

## Acknowledgments

The authors would like to thank the participants for their time and cooperation throughout the study.

## Author contributions

**Conceptualization:** Scott Newbould, Josh Walker, Alexander J Dinsdale, Sarah Whitehead, Gareth Nicholson.

**Formal analysis:** Scott Newbould.

**Investigation:** Scott Newbould.

**Methodology:** Scott Newbould, Josh Walker, Alexander J Dinsdale, Sarah Whitehead, Gareth Nicholson.

**Project administration:** Scott Newbould.

**Supervision:** Gareth Nicholson.

**Visualization:** Josh Walker.

**Writing – original draft:** Scott Newbould.

**Writing – review & editing:** Josh Walker, Alexander J Dinsdale, Sarah Whitehead, Gareth Nicholson.

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
