## [Decision Letter · Decision Letter 0]

24 Jul 2024

Dear Dr. Newbould,

Thank you for submitting your manuscript to PLOS ONE. After careful consideration, we feel that it has merit but does not fully meet PLOS ONE’s publication criteria as it currently stands. Therefore, we invite you to submit a revised version of the manuscript that addresses the points raised during the review process.

We look forward to receiving your revised manuscript.

Kind regards,

Charlie M. Waugh

Academic Editor

PLOS ONE

Journal Requirements:

Reviewers' comments:

Reviewer's Responses to Questions

**Comments to the Author**

1. Is the manuscript technically sound, and do the data support the conclusions?

Reviewer #1: Yes

Reviewer #2: Yes

2. Has the statistical analysis been performed appropriately and rigorously?

Reviewer #1: Yes

Reviewer #2: Yes

3. Have the authors made all data underlying the findings in their manuscript fully available?

Reviewer #1: Yes

Reviewer #2: Yes

4. Is the manuscript presented in an intelligible fashion and written in standard English?

Reviewer #1: Yes

Reviewer #2: Yes

Reviewer #1: This study provides new female dataset for various global performance measures (i.e., CMJ jump height, mid-thigh pulls metrics) and other quantifiable muscle-tendon characteristics, which expands the scientific understanding of test-retest reliability in these performance metrics in the given population. The introduction, discussion, and conclusion of this manuscript were well-written, so most of my review are on the technical sides (method). Please see below points raised:

Line 132

The ultrasound image was collected at 15 Hz, which seems to be a lower frequency. Could the authors justify why 15 Hz was chosen in this study, and not 30 Hz or above ? Did the authors compared similar isometric plantarflexions to other frame rates to ensure 15 Hz was sufficient? This could be one of the reasons why traditional stiffness results were less reliable in this study because there were less data points to fit a curve over a specified region to calculate these traditional stiffness values (and hence a less accurate curve was fitted).

Line 161-162

The authors mentioned “determined by the greatest flight time” to define the best three trials of the jump. However, throughout the method section, the authors did not explicitly define how jump height was calculated. The authors mentioned a software (Line 157) to make these calculations, but in my opinion, this was not particularly clear. Looking at Table 2, Table 3, and the supplementary file, the reported jump height included both the flight time-based calculation and the velocity-based calculation. It would be a good practice to explain clearly which method was used in this manuscript to determine test-retest reliability. Usually, velocity-based calculation is the “impulse method”, where the net vertical impulse of the jump during the entire ground contact phase was used to calculate change in COM velocity at take-off. Impulse method is generally more reliable than flight time method since it is not affected by landing posture. The authors addressed the limitation of flight time method in the discussion (Line 321), but please define the calculation method earlier in the method section. Also, please consider analysing velocity-derived jump height results if this has not been done.

Line 163-164

This statement is mostly correct, but the term “isolate the lower body” might seem less accurate. This sentence might improve better if we change it to “remove the effect of arm swing to ensure jump contribution was predominantly lower limb driven” or something similar. The reason is, even without arm swing, the torso/trunk can still add to the final jump performance and therefore we cannot really isolate lower body, unless we perform jumps on a Smith machine that restricts trunk contribution.

Line 168-170

For the SJ protocol, the authors did not explain whether the “self-selected depth” for each participant was strictly controlled across different trials and/or different sessions. Unlike CMJ, the initial position of a SJ could change more easily than CMJ if not monitored carefully, making the SJ results a function of depth manipulation. Participants might vary their depth selection between trials or sessions, which might contribute to some variability in the data. The data reported in this study showed a high reliability for SJ jump height, but if this outcome was based on a variable squat depth, then our interpretation of SJ jump height would be more complicated. Please elaborate further on whether self-selected depth for SJ was controlled in the method section. If not controlled, it would be good to briefly mention in the discussion as a limitation.

Line 261-276

This paragraph is well-written. One extra point to consider is that traditional stiffness calculation can (not always) involve fitting a curve to the raw data, and therefore the stiffness values can be affected hugely by the shape of the fitted curve. On the other hand, the method adopted in this study divided the maximal AT force by the maximal strain, which did not require curve fitting and reduced the possibility of errors associated with curve fitting. This is just a comment, so the authors can decide whether to agree with this comment or not.

Reviewer #2: General comments:

The current study aimed to test-retest reliability of local and global MTU function assessments during the early follicular phase of the menstrual cycle. The topic is highly valuable to the scientific community as research with females is largely neglected in MTU both in cross-sectional but even more specifically in interventional studies. The manuscript is generally well-written, but includes several points to improve prior being fit for publication. Please see my comments / suggestions below.

Specific comments:

Title and the whole manuscript (starting for instance L48): Terms “local and global MTU function” are not quite correct. Function does not equal mechanical properties. Mechanical properties like tendon strain or stiffness are quantities that describe material characteristics, which can affect function. In the case of the tendon, it’s function is to transmit forces and store/release energy and this is influenced by its mechanical properties. I would suggest to use “local and global MTU assessment“, then you are on a correct side. Double check the whole manuscript if the term “function” is used correctly.

L58-L60: Biological variation in what exactly?

L73: Change “jump” to “jumping”

L76-L81: It is quite hard to compare those studies due to the methodological differences (i.e. different joint configurations, accounting for inevitable joint changes to measured elongation, the region used for calculating tendon stiffness, differences in loading protocol), which explains largely the variation in the reliability. Therefore I would be careful with the statement of “mixed findings” because the least controlled and less favorable conditions tend to lead to a lower reliability / higher variability. Pooling these results in one pot is not a wise and legitimate way to interpret the findings from these studies. I think the authors should recognize the reasons why this cannot be done.

L99: “..., whilst controlling for menstrual cycle phase.” is a weird way to say it, because you are not really controlling it rather than performing the measurements within a specific phase of the menstrual cycle.

L123: “of AT mechanical function” is not correct. Change it to ” mechanical properties”.

L128: I think a brief explanation of the Barfod et al. 2015 method using US to measure tendon length would be helpful for the readers, because it is not really a conventional measure across different studies and research groups. Mention that it takes into account the curvature of the AT.

L144: Did you also account for the axis misalignment between the ankle joint centre and the dynamometer’s axis of rotation? If not this can affect directly the plantarflexion MVCs. Considering the inevitable ankle joint rotation seemed to be more than 10deg, the peak joint moments would have been produced in a less optimal joint configuration.

L149: “To investigate the reliability of the manual measurement of AT elongation, a random sample of 10 maximal contractions were selected from the whole dataset.” I do not see the reasoning for not including all the maximal contractions from all the participants for the reliability analysis. This could potentially artificially increase the intra- and inter-rater reliability.

L126-L154: Did I miss it or you have forgotten to describe how you measured the absolute tendon stiffness and normalized tendon stiffness? Because in the results you are demonstrating the ICC values etc. Which region of the force-elongation relationship was used for calculating the slope etc.

L177-L190: Did you take into account the inevitable axis misalignment between the isokinetic device and the knee joint centre? If not this can directly affect your outcomes and needs to be mentioned here and in the limitations.

L183-L186: How were the joint configurations determined? Motion capture? During the movement there are definitely inevitable changes in the joint angles. Does and how much could this affect the resultant measured peak GRF. Only the left leg stood on the force plate? I think needs to be mentioned. And if not, then why?

Table 2, 3, 4 and 5: You have not described how you calculated most of the parameters, just referred to a previous study. Surely some of them need explaining.

L259 and L373: “blunt instruments” I would refrain from using this term in the manuscript.

L438: In the reference Robshaw DC, you are missing the full information about this PhD Thesis.

**Do you want your identity to be public for this peer review?** For information about this choice, including consent withdrawal, please see our Privacy Policy

Reviewer #1: **Yes: ** Eric Yung-Sheng Su

Reviewer #2: No

---

## [Author Response · Author response to Decision Letter 1]

7 Aug 2024

All comments have been responded to in the 'Response to Reviewers' letter.

---

## [Decision Letter · Decision Letter 1]

22 Oct 2024

Between-day reliability of local and global muscle-tendon unit assessments in female athletes whilst standardising menstrual cycle phase

PONE-D-24-24939R1

Dear Dr. Newbould,

We’re pleased to inform you that your manuscript has been judged scientifically suitable for publication and will be formally accepted for publication once it meets all outstanding technical requirements.

Kind regards,

Laura-Anne Marie Furlong

Academic Editor

PLOS ONE

Additional Editor Comments (optional):

Reviewers' comments:

Reviewer's Responses to Questions

**Comments to the Author**

Reviewer #2: All comments have been addressed

2. Is the manuscript technically sound, and do the data support the conclusions?

Reviewer #2: Yes

3. Has the statistical analysis been performed appropriately and rigorously?

Reviewer #2: Yes

4. Have the authors made all data underlying the findings in their manuscript fully available?

Reviewer #2: Yes

5. Is the manuscript presented in an intelligible fashion and written in standard English?

Reviewer #2: Yes

Reviewer #2: The authors have responded to my comments / concerns and the quality of the manuscript has been now significantly improved. I have no further issues and would like to congratulate the authors for a interesting and timely manuscript.

**Do you want your identity to be public for this peer review?** For information about this choice, including consent withdrawal, please see our Privacy Policy

Reviewer #2: No

---

## [Editor Report · Acceptance letter]

PONE-D-24-24939R1

PLOS ONE

Dear Dr. Newbould,

I'm pleased to inform you that your manuscript has been deemed suitable for publication in PLOS ONE. Congratulations! Your manuscript is now being handed over to our production team.

Kind regards,

on behalf of

Dr. Laura-Anne Marie Furlong

Academic Editor

PLOS ONE